# PaRaDe: Passage Ranking using Demonstrations with Large Language Models

Andrew Drozdov♠◇∗    Honglei Zhuang♠    Zhuyun Dai♠    Zhen Qin♠
Razieh Rahimi◇    Xuanhui Wang♠    Dana Alon♠    Mohit Iyyer◇
Andrew McCallum◇    Donald Metzler♠    Kai Hui♠†

♠Google    ◇UMass Amherst CICS

## Abstract

Recent studies show that large language models (LLMs) can be instructed to effectively perform *zero-shot* passage re-ranking, in which the results of a first stage retrieval method, such as BM25, are rated and reordered to improve relevance. In this work, we improve LLM-based re-ranking by algorithmically selecting *few-shot* demonstrations to include in the prompt. Our analysis investigates the conditions where demonstrations are most helpful, and shows that adding even one demonstration is significantly beneficial. We propose a novel demonstration selection strategy based on difficulty rather than the commonly used semantic similarity. Furthermore, we find that demonstrations helpful for ranking are also effective at question generation. We hope our work will spur more principled research into question generation and passage ranking.

## 1 Introduction

Large language models (LLMs) exhibit strong performance on a variety of tasks without additional task-specific fine-tuning. Their success is often attributed to *in-context learning*, where the parameters of the language model are frozen and it learns how to perform a new task by reading demonstrations in the prompt (Brown et al., 2020; Basu et al., 2023; Min et al., 2022; Akyürek et al., 2023).

While LLMs are often used to *generate* answers, our focus is on *scoring* for the task of passage re-ranking—passages are first retrieved by an efficient retriever, e.g. BM25, then rated and reordered by the LLM. Existing works like UPR (Sachan et al., 2022) demonstrate promising results for *zero-shot* ranking using LLM. We aim to improve over zero-shot ranking by including demonstrations in the prompt and explore multiple strategies for selecting demonstrations. Manual selection is often sub-

optimal and requires a human-in-the-loop when using the LLM for a new task. Instead, we seek a method that finds effective demonstrations automatically, with minimal or no human involvement.

In this paper, we investigate approaches for automatic demonstration selection to improve upon UPR's zero-shot ranking approach. Our initial analysis highlights the complex nature of the problem, showing that ranking performance varies drastically depending on the demonstrations included in the prompt. Furthermore, simply including more demonstrations does not always lead to better ranking quality. Next, we investigate the use of established demonstration selection methods, i.e. similarity-based selection (Rubin et al., 2022; Luo et al., 2023), on ranking tasks and show that similarity of demonstrations does not correlate well with ranking quality. Thereafter, we propose difficulty-based selection (DBS) as a simple and effective approach to automatically find challenging, i.e. low likelihood, demonstrations to include in the prompt. Although we prompt frozen LLMs, we intend to emulate the training dynamics of fine-tuning, and choose hard samples because they potentially correspond to large gradient updates and are often chosen to improve learning in gradient descent (Shrivastava et al., 2016; Chang et al., 2017). Finally, given the increasing importance of question generation for ranking (Nogueira et al., 2019; Bonifacio et al., 2022; Dai et al., 2023; Jeronymo et al., 2023), we extend the uses of the proposed difficulty-based selection for better question generation.

To this end, we present Passage Ranking with Demonstrations (PaRaDe). Our main contributions include: (1) analysis highlighting the complexity of demonstration selection; (2) DBS, an automatic and effective way to choose demonstrations; and (3) extensive experiments on re-ranking and question generation, including results with an extension of DBS that jointly selects multiple demonstrations.

---

∗Work completed while a Student Researcher at Google.
†Final author. Also generated Table 1 results using DQL-scored demonstration candidates from AD.

## 2 LLM Re-ranking by Query Likelihood

**Background.** Given a query $q$ and set of initially retrieved documents $D$, the goal is to rank each document $d$ in $D$ by relevance with respect to $q$. UPR (Sachan et al., 2022) introduces a zero-shot method to rank documents according to the log-likelihood of $q$ given $d$ using a large language model,

$$\ell(q \mid d) \propto \sum_{i=1..N} \log P(q_i \mid d, q_{1:i-1}), \quad (1)$$

which resembles the query likelihood (QL) retrieval model (Ponte and Croft, 1998). It is factorized using the probabilities of each token in the query $q_i$ and prefix of that token $q_{1:i-1}$. Extending QL to include demonstrations in the context yields,

$$\ell(q \mid d) \propto \sum_{i=1..N} \log P(q_i \mid z_1, ..., z_k, d, q_{1:i-1}),$$
$$(2)$$

where each $z_i$ is a positive query-document pair.

## 3 Experiments on TREC and BEIR

To empirically measure the effectiveness of demonstration selection, we conduct analysis using the re-ranking task on TREC 2019 and 2020, and further perform evaluation on seven datasets from BEIR (Thakur et al., 2021). We use language models known to effectively incorporate demonstrations through in-context learning, including the Flan-T5-XL and XXL (Chung et al., 2022) and the more recently PaLM 2-S (Google et al., 2023).

**Setup.** For each dataset, we retrieve the top-100 documents using BM25 from Pyserini (Lin et al., 2021). The LLMs re-rank these top documents using query likelihood (§2) in a point-wise manner, and the re-ranked results are evaluated using nDCG@10. Herein, the instruction (Table 4) and the selected demonstrations composite the prompt string when scoring each (query, passage) pair.

## 4 Demonstrations for Ranking

### 4.1 The Impact of Demonstrations

In this section, we investigate the helpfulness of demonstrations for ranking and sensitivity to the choice of demonstration. We explore multiple strategies for selecting demonstrations: random sampling, similarity-based selection (SBS), and our new approach difficulty-based selection (DBS). Our findings indicate that the choice of demonstrations considerably impacts ranking performance.

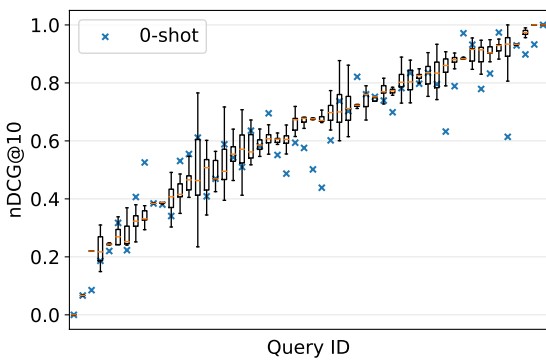

Figure 1: Statistics for nDCG@10 on TREC 2020, aggregated using the same query with 100 different one-shot demonstrations. Flan-T5-XXL is used for re-ranking. Zero-shot results included for reference.

**Demonstrations influence ranking.** Figure 1 shows ranking performance of zero-shot and one-shot prompts, and we can see that LLMs are quite sensitive to the choice of demonstrations. We randomly sampled 100 demonstrations for use in one-shot re-ranking with query likelihood and also compare against zero-shot. On 25.9% of queries the minimum one-shot nDCG@10 outperforms zero-shot, while zero-shot outperforms the max one-shot for 11.1% of queries. It is worth emphasizing that there is a high variance across different one-shot demonstrations on many queries.

**Increasing demonstrations does not necessarily help.** Surprisingly, we find little benefit when randomly sampling more demonstrations beyond one-shot (see Figure 2). There is a minor, almost negligible improvement in median performance when using four demonstrations, and even less change with eight. We do notice slightly decreased variation as we increase the number of demonstrations. This highlights the difficulties in selecting demonstrations for ranking tasks beyond one-shot.

Given the large variance in one-shot performance and the difficulty to improve performance by increasing demonstrations, we need an effective way to select high performing demonstrations.

**Similarity-based selection is limited.** A simple and widely used baseline for selecting demonstrations is semantic similarity (Rubin et al., 2022; Luo et al., 2023). Intuitively, it makes sense that semantically similar demonstrations to the test query would help teach the LLM how to re-rank, although, if the LLM is already familiar with the demonstrations then it is not clear whether they

will prove helpful. We perform post-hoc analysis on our TREC 2020 experiments with 100 random one-shot demonstrations. We measured semantic similarity using cosine similarity and Universal Sentence Encoder (Cer et al., 2018) embeddings of the demonstration and test queries. By comparing the semantic similarity and the nDCG, we ascertain there is little or no correlation between high semantic similarity and strong re-ranking. Our findings show this correlation is significant only 5% of the time, thus conclude that similarity alone has clear limitations for demonstration selection. In the next subsection, we explore a new technique inspired by in-context learning dynamics rather than semantic similarity for selecting demonstrations.

## 4.2 Difficulty-based Selection (DBS)

We propose difficulty-based selection to find challenging demonstrations to include in the prompt. We estimate difficulty using demonstration query likelihood (DQL):

$$DQL(z) \propto \frac{1}{|q^{(z)}|} \log P(q^{(z)} \mid d^{(z)}),$$

then select the demonstrations with the lowest DQL. Intuitively, this should find hard samples that potentially correspond to large gradients had we directly trained the model instead of prompting.[1]

### 4.2.1 Difficult Demonstrations are Beneficial

On TREC 2020, we observe a statistically significant (p=0.008) correlation (0.26) between negative DQL and ranking performance with the 100 random one-shot demonstrations from §4.1. The lowest DQL outperforms average nDCG@10 across the 100 random demonstrations (64.1 vs. 62.8).

To further investigate how well DQL works for selecting demonstrations, we sample easy or hard demonstrations from the full MS Marco training data rather than only using our initially selected 100. We form four bins by sampling 30 demonstrations from each of the bottom-1% and 10% by DQL (these are the hardest, and should give the best results), and the same with easy ones. We plot the mean and max nDCG@10 for each bin in Figure 3. For Flan-T5-XXL, the performance improves as we use more challenging demonstrations. This trend is less prominent for Flan-T5-XL.

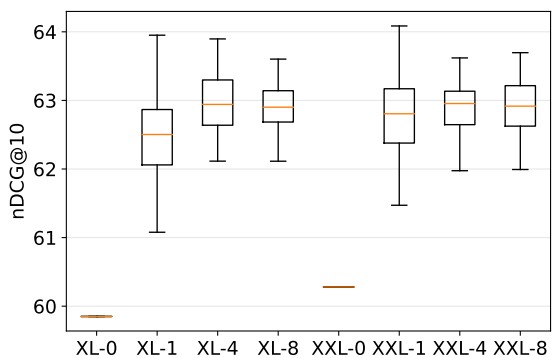

Figure 2: Statistics for nDCG@10 on TREC 2020, aggregated using 100 different $k$-shot demonstrations with Flan-T5-XL and XXL models. Number of demonstrations ($k$) shown after the dash.

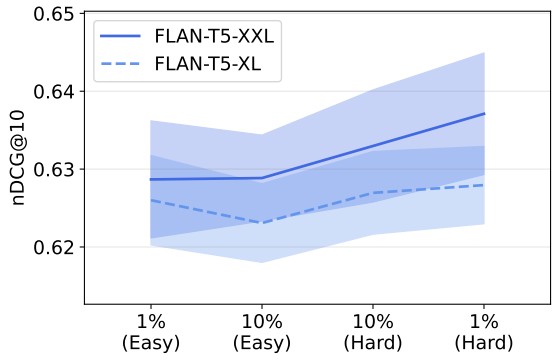

Figure 3: The nDCG@10 of DQL-based bins measured on TREC 2020. The x-axis increases in difficulty of demonstration from left-to-right.

## 5 Main Results and Discussion

### 5.1 DBS for TREC and BEIR

In Table 1, we compare DBS with zero-shot and manual demonstration selection. Manual curation is with demonstrations from Promptagator (Dai et al., 2023), which uses up to eight demonstrations depending on the task.[2] The results show that demonstrations often improve re-ranking on TREC and BEIR, and furthermore, that our DBS is effective for automatic demonstration selection. The improvement over zero-shot can be substantial, such as the case for TREC 2020, FiQA, and NQ where using demonstrations leads to more than 3-points improvement in all settings. When zero-shot outperforms few-shot, it is only by a small margin and often on datasets that require complex retrieval such as FEVER or HotpotQA. DBS outperforms

---

[1]Recent theories on the effectiveness of in-context learning view few-shot prompting similarly to fine-tuning on the demonstrations in the prompt (Basu et al., 2023).

[2]For example, there are eight, six, and two demonstrations for TREC, FiQA, and Scifact.

| | T19 | T20 | FiQA | Scifact | BioASQ | FEVER | HotpotQA | NQ | Quora |
|---|---|---|---|---|---|---|---|---|---|
| BM25 | 50.60 | 48.00 | 23.60 | 66.50 | 46.45 | 75.32 | 60.27 | 32.84 | 78.83 |
| **Flan-T5-XL** | | | | | | | | | |
| 0-shot | **61.10** | 59.90 | 38.20 | 70.40 | 54.34 | 68.02 | **72.79** | 40.26 | 77.09 |
| Promptagator | 61.00 | 61.40 | 43.40 | 71.90 | 54.57 | 69.40 | 72.36 | **44.70** | **84.53** |
| DBS 1-shot | 59.80 | 62.50 | 44.10 | 72.00 | **54.81** | 69.42 | 72.69 | 44.07 | 83.66 |
| DBS 4-shot | 61.00 | **63.00** | **44.70** | **72.70** | 54.32 | **70.46** | 72.53 | 44.58 | 84.32 |
| **Flan-T5-XXL** | | | | | | | | | |
| 0-shot | 61.80 | 60.30 | 42.90 | 73.00 | 55.11 | **78.17** | 72.56 | 44.93 | 83.70 |
| Promptagator | 61.90 | 63.30 | 47.40 | 73.80 | 55.32 | 78.00 | 73.53 | 47.90 | 85.56 |
| DBS 1-shot | 62.66 | **63.99** | 47.60 | 74.30 | 55.41 | 77.62 | **74.11** | **48.46** | 85.31 |
| DBS 4-shot | **63.38** | 62.93 | **47.70** | **74.50** | **55.71** | 77.68 | 73.78 | 48.41 | **85.73** |
| **Palm 2-S** | | | | | | | | | |
| 0-shot | 55.84 | 55.55 | 38.26 | 74.69 | 52.31 | 76.94 | 71.98 | 43.33 | 83.51 |
| Promptagator | 61.24 | 60.92 | **48.11** | **76.89** | **55.69** | 78.18 | **75.43** | 44.71 | **85.89** |
| DBS 1-shot | 58.50 | 60.62 | 46.99 | 74.87 | 53.43 | 78.07 | 72.93 | 42.91 | 85.52 |
| DBS 4-shot | **61.39** | **61.20** | 47.96 | 76.52 | 54.34 | **78.59** | 75.36 | **49.84** | 85.81 |

Table 1: nDCG@10 for TREC 2019/2020 and seven BEIR datasets after re-ranking the top-100 documents retrieved by BM25. Flan-T5-XL/XXL and Palm 2-S perform prompt-based re-ranking, where zero-shot is identical to UPR (Sachan et al., 2022). We also use manually selected demonstrations from Promptagator (Dai et al., 2023).

Promptagator demonstrations in many settings, including by a large margin for TREC 2019 with Flan-T5-XXL and NQ with Palm 2-S.

**Demonstration filtering.** When using DBS, we return the top-30 demonstrations and then perform a lightweight manual filtering to remove demonstrations with incorrect labels.[3] We also remove duplicate queries. In the future, filtering for incorrect labels should be easy to automate, and it would be interesting to explore how DBS can be used to mine for incorrect annotations.

### 5.2 Comparison to Random Selection

Our findings thus far show that *zero-shot* ranking is outperformed by demonstration-based ranking with DBS in almost every setting. To understand how much potential there is to further improve demonstration selection, we compare DBS using Flan-T5-XXL against 10 randomly selected one-shot demonstrations for TREC and BEIR (see Appendix A.4 for full results). We see that the max nDCG from random selection outperforms DBS in five out of nine datasets. This suggests LLM-based ranking may be further improved through

advanced selection. In the future it would be helpful to see a richer distribution of performance using substantially more than 10 random demonstrations.

### 5.3 DBS with Conditional DQL (CDQL)

DQL overlooks that variations in demonstration difficulty depends on the other demonstrations in the context. To jointly consider the difficulty of all demonstrations in the prompt, we propose *conditional* demonstration query likelihood (CDQL):

$$CDQL(z_1, z_2, ..., z_K) \propto$$
$$\sum_{i=1..K} \frac{1}{|q^{(z_i)}|} \log P(q^{(z_i)} \mid z_{1:i-1}, d^{(z_i)})$$

In preliminary results, we find that Flan-T5-XXL with CDQL improves over DQL on TREC 2019 and 2020, respectively giving 63.5 vs. 63.4 and 64.4 vs. 64.0 nDCG.[4] To use CDQL we chose 30 demonstrations first by DQL, filtered for any incorrect labels, then computed CDQL for each permutation including four demonstrations and took the lowest CDQL. We leave further exploration of CDQL to future work, and believe it may be beneficial when selecting more than one demonstration.

---

[3]Statistics for filtering are in Appendix A.2.

[4]Results for CDQL are in Table 3, in the Appendix.

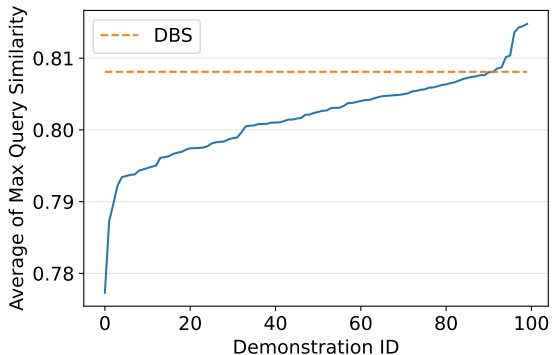

Figure 4: For each demonstration, we compute the semantic similarity between ground-truth and the synthetic queries. We first measure the max similarity by demonstration and query. Then we average this across all queries, giving a single scalar per demonstration. The dashed line shows the "average max similarity" for the demonstration chosen using DBS.

## 5.4 DBS for Question Generation

As an auxiliary evaluation of DBS we study question generation, which plays important roles in different NLP applications (Dai et al., 2023; Bonifacio et al., 2022; Jeronymo et al., 2023; Nogueira et al., 2019; Ma et al., 2021). Using the top-100 passages retrieved from BM25 for each query in TREC 2020, we greedily generate with Flan-T5-XXL 100 questions per passage using a one-shot prompt and the 100 random demonstrations from §2. We compare the generated questions from random demonstrations and the ones from DBS (1-shot). For each query, we compute the maximum cosine similarity between the ground truth and generated questions after embedding with Universal Sentence Encoder (Cer et al., 2018). The average of the max similarity among 100 random demonstrations is 0.8018 (min=0.7770 and max=0.8150), whereas, we achieve 0.8081 when using DBS (one-shot). Compared with the random demonstrations, the DBS result ranks 8% highest similarity in the population and is significantly greater than the mean (p=4e-18) according to two-tailed t-test (Figure 4). These findings indicate that the demonstrations effective for LLM-based *scoring* of passages are similarly effective for *generation* of questions.

## 6 Related Work

Concurrent with UPR, PromptRank (Khalifa et al., 2023) is the most related prior work, using demonstrations to re-rank "document paths" for multihop-QA. Details of how they select demonstrations is unclear, motivating us to conduct our own study.

Our difficulty-based demonstration selection (§4.2) is closely related to active learning (Dagan and Argamon, 1995; Roy and McCallum, 2001; Settles, 2009). Similarly, Diao et al. (2023) measure uncertainty with generation instead of scoring. Zhang et al. (2022) formulate demonstration selection as a reinforcement learning problem. Rubin et al. (2022) use LLM-scoring to find hard negatives for their trained demonstration retriever. Others explore demonstration ordering (Lu et al., 2022) and joint selection (Drozdov et al., 2023; Levy et al., 2023; Agrawal et al., 2023; Ye et al., 2023). Concurrent to our work, Li and Qiu (2023) perform multiple rounds of hill climbing to find groups of demonstrations that perform well according to a validation set. In contrast, DBS selects demonstrations directly and does not rely on validation.

Discriminative methods are widely used in supervised ranking (Zhuang et al., 2023; Nogueira dos Santos et al., 2020; Hui et al., 2022). Listwise prompting is an alternative to query likelihood, but requires a sliding window strategy as not all documents fit in the context (Ma et al., 2023; Sun et al., 2023). Rather than query likelihood, HyDE (Gao et al., 2023) achieves zero-shot ranking through document generation, which we hypothesize would be improved through demonstrations. PaRaDe is bounded by the first stage BM25 retrieval, and it may be fruitful to explore approaches that align first stage retrieval with our demonstration-based approach (Yadav et al., 2022).

## 7 Conclusion

In this work we present Passage Ranking with Demonstrations (PaRaDe), an extensive study on the topic of using demonstrations to improve re-ranking performance of LLMs. We show the challenges of applying demonstrations effectively, and that performance heavily relies on selecting "good" demonstrations. We propose a simple yet effective selection method, named difficulty-based selection (DBS), and confirm its effectiveness in both re-ranking using query likelihood scoring and query generation tasks. For future work, we plan to combine difficulty-based selection with similarity-based selection as an effort to further improve the robustness and effectiveness of the selected demonstrations, and extend DBS to other ranking paradigms (Qin et al., 2023; Sun et al., 2023).

## Acknowledgements

We thank Tal Schuster for their detailed comments on earlier versions of this manuscript. We are grateful to Vinh Tran for insightful discussions regarding the Flan family of models and their in-context learning capabilities.

## Limitations

One limitation of DBS is when naively used to select multiple demonstrations, the demonstrations that may appear challenging at first may become relatively easy once the other demonstrations have been processed in the prompt context. We partially address this by replacing DQL in DBS with CDQL (§5.3) so that demonstrations are selected jointly rather than scored individually. Our CDQL approach selects a high scoring subset from the initial list provided by DQL. More challenging combinations of demonstrations may be available by searching the entire candidate set, but exact search is computationally prohibitive. Another limitation is that we only incorporate positive demonstrations, and for retrieval, training with hard negatives is often beneficial to model performance. We hypothesize a DBS-like algorithm can be used to find hard negatives, but it may be important to add signals distinguishing positive from negative demonstration when using LLMs with query likelihood for ranking.

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

# A  Appendix

## A.1  Prompt Format

The instructions we use are in Table 4. We also include a short prefix indicating the start of document or query, shown in Table 5. These are used once for the test query and document, and duplicated for any demonstrations in the prompt. For few-shot the prompt includes an instruction, one or more demonstrations, and the test data. Scores for ranking are computed only on the query. For zero-shot, no demonstrations are included.

## A.2  Statistics for Demonstration Filtering

Table 2 shows statistics for demonstration filtering, including the ranks of selected demonstrations and the number of demonstrations that are skipped per dataset. Demonstrations are skipped if they are incorrectly labeled or duplicates of already selected demonstrations. The demonstrations are selected from the training data, and MSMarco is used for TREC 2019 and 2020. In the case of FEVER, we needed to truncate long demonstrations so they would fit in the prompt. Sometimes this would inadvertently make labels incorrect as necessary information resided in the removed text. We always perform any preprocessing and truncation before running demonstration selection.

## A.3  Results with CDQL

Table 3 includes results using conditional demonstration query likelihood (CDQL). These are only preliminary results, and we believe that CDQL should be an effective alternative to DQL when selecting more than one demonstration.

## A.4  Results with Random Selection

In Table 6 we compare DBS one-shot with random one-shot demonstrations when using Flan-T5-XXL. Random performance is aggregated across 10 randomly selected demonstrations, and we show the mean, standard deviation, minimum, and maximum values. The results indicate there is further opportunity to improve demonstration selection, and also that demonstration selection is a challenging task.

| Dataset | Ranks of Selected | No. Skipped |
|---------|-------------------|-------------|
| MSMarco | 1,2,3,4 | 0 |
| FiQA | 4,5,6,7 | 3 |
| Scifact | 5,6,7,8 | 3 |
| BioASQ | 1,3,5,18 | 5 |
| FEVER | 5,8,16,22 | 17 |
| HotpotQA | 2,3,6,8 | 4 |
| NQ | 3,4,5,8 | 4 |
| Quora | 1,2,4,5 | 1 |

Table 2: Statistics for demonstration filtering.

| | T19 | T20 |
|---|---|---|
| BM25 | 50.6 | 48.0 |
| **Flan-T5-XXL** | | |
| 0-shot (UPR) | 61.8 | 60.3 |
| Promptagator | 61.9 | 63.3 |
| DBS 1-shot (DQL) | 62.7 | **64.0** |
| DBS 4-shot (DQL) | **63.4** | 62.9 |
| DBS 4-shot (CDQL) | **63.5** | **64.4** |

Table 3: nDCG@10 on TREC 2019 and 2020 when using Flan-T5-XXL. We compare zero demonstrations, manual curation (Promptagator), and automatic selection with DBS using DQL or CDQL.

| Dataset | Instruction |
|---|---|
| TREC 2019 | [web] I will check whether what you said could answer my question. |
| TREC 2020 | [web] I will check whether what you said could answer my question. |
| BEIR FiQA | [web] I will check if what you said could verify my question. |
| BEIR Scifact | [web] I will check if the argument you said could verify my scientific claim. |

Table 4: The instructions for zero-shot and few-shot prompts. Zero-shot prompts include only the instruction and test document, with scoring on the test query. Few-shot prompts also include demonstrations. The same prompts are used for both query likelihood and question generation, although question generation excludes the test query.

| Dataset | Query-Document Template |
|---|---|
| TREC 2019 | You said: DOCUMENT <newline> I googled: QUERY |
| TREC 2020 | You said: DOCUMENT <newline> I googled: QUERY |
| BEIR FiQA | You said: DOCUMENT <newline> I googled: QUERY |
| BEIR Scifact | Argument: DOCUMENT <newline> My scientific claim: QUERY |

Table 5: The prompt template for zero-shot and few-shot prompts. Zero-shot prompts include only the instruction and test document, with scoring on the test query. Few-shot prompts also include demonstrations. The same prompts are used for both query likelihood and question generation, although question generation excludes the test query.

|  | T19 | T20 | FiQA | Scifact | BioASQ | Fever | HotpotQA | NQ | Quora |
|---|---|---|---|---|---|---|---|---|---|
| BM25 | 50.58 | 47.96 | 23.61 | 66.47 | 46.45 | 75.32 | 60.27 | 32.84 | 78.83 |
| 0-shot | 61.80 | 60.30 | 42.90 | 73.00 | 55.11 | **78.17** | 72.56 | 44.93 | 83.70 |
| Promptagator | 61.90 | 63.30 | 47.40 | 73.80 | 55.32 | 78.00 | 73.53 | 47.90 | 85.56 |
| DBS 1-shot | 62.66 | **63.99** | 47.60 | 74.30 | 55.41 | 77.62 | **74.11** | **48.46** | 85.31 |
| DBS 4-shot | **63.38** | 62.93 | **47.70** | **74.50** | **55.71** | 77.68 | 73.78 | 48.41 | **85.73** |
| DBS 1-shot | **62.66** | **63.99** | 47.60 | 74.30 | 55.41 | 77.62 | 74.11 | 48.46 | 85.31 |
| R 1-shot (avg) | 62.02 | 62.84 | **48.58** | **75.58** | **55.54** | **80.41** | 71.98 | **49.28** | 84.71 |
| R 1-shot (std) | 0.47 | 0.44 | 0.27 | 0.37 | 0.23 | 1.39 | 0.24 | 0.38 | 0.35 |
| R 1-shot (min) | 60.92 | 62.19 | 48.05 | 74.97 | 55.34 | 77.64 | 71.56 | 48.61 | 83.78 |
| R 1-shot (max) | 62.80 | 63.50 | 48.93 | 76.21 | 56.03 | 81.93 | 72.31 | 49.71 | 85.05 |

Table 6: nDCG@10 on BEIR datasets, using all queries when using Flan-T5-XXL. We compare DBS 1-shot with random (R) 1-shot. Random performance is aggregated across 10 randomly selected demonstrations, and we show the mean, standard deviation, minimum, and maximum values. DBS 1-shot is underlined if it is greater than the maximum random 1-shot and vice versa. When maximum random 1-shot outperforms DBS 1-shot, this suggests there is further opportunity to improve demonstration selection, and also that demonstration selection is a challenging task.