# OpenReview forum: "PaRaDe: Passage Ranking using Demonstrations with LLMs"
_EMNLP/2023/Conference — EMNLP 2023 Findings_

### Official Review · Reviewer_E2cL · 2023-07-30

**Soundness:** 4

**Excitement:**

3: Ambivalent: It has merits (e.g., it reports state-of-the-art results, the idea is nice), but there are key weaknesses (e.g., it describes incremental work), and it can significantly benefit from another round of revision. However, I won't object to accepting it if my co-reviewers champion it.

**Paper Topic And Main Contributions:**

This paper proposes a new LLM for passage ranking method called DQL, which is proven useful through a series of experiments.

**Questions For The Authors:**

What the formal definition of DQL, how is it calculated?

**Reasons To Accept:**

1. The proposed technique is simple and effective.
2. The experiments on the demonstration in Section 3.1 is helpful for future research.

**Reasons To Reject:**

1. No formal definition of DQL is given in the passage. Though I have a rough understanding of DQL, I cannot understand the detailed calculation process. A detailed formula should be elaborated.
2. The effectiveness is verified in limited LLM and datasets, which may not be convincing.

**Reproducibility:**

3: Could reproduce the results with some difficulty. The settings of parameters are underspecified or subjectively determined; the training/evaluation data are not widely available.

**Reviewer Confidence:**

3: Pretty sure, but there's a chance I missed something. Although I have a good feel for this area in general, I did not carefully check the paper's details, e.g., the math, experimental design, or novelty.

---

> ### Author Rebuttal · Authors · 2023-08-29
>
> Thank you for your thoughtful comments. We're glad to hear that our work is considered a simple and effective approach (E2cL,auEx) that improves a previous method (auEx), is helpful for future research (E2cL,auEx) and well motivated (auEx), and focuses on an interesting research question that pushes the boundaries of LLM capabilities (99xq). We respond to the questions from the review below.
>
>
> **Reason to Reject 1. Formal definition of DQL.**
>
> Thank you for mentioning this. Demonstration query likelihood is similar to query likelihood although applied to demonstrations instead of test queries. The only difference is that we divide by the number of tokens, since queries across demonstrations may have different length. For a (query, passage) pair from the training dataset, the formulation is as follows:
>
> $DQL(q \mid d) \propto \frac{1}{N} \sum_{i=1..N} \log P (q_i | d, q_{0:i-1})$
>
>
> **Reason to Reject 2. The effectiveness is verified in limited LLM and datasets, which may not be convincing.**
>
> We have run experiments on an additional six datasets and also include more recent LLM, namely, Palm for comparisons. It can be seen that the conclusions still hold on these additional dataset and when using a more recent LLM. DBS 4-shot provides a similar boost as with Promptagator k-shot. Note that up to eight demonstrations are used in Promptagator kshots.
>
> |   | T19 | T20 | FiQA | Scifact | BioASQ | FEVER | HotpotQA | NQ | Quora |
> |---------|-----|-----|------|---------|--------|-------|----------|----|-------|
> | BM25 | 50.60 | 48.00 | 23.60 | 66.50 | 46.45 | 75.32 | 60.27 | 32.84 | 78.83 |
> |---------|-----|-----|------|---------|--------|-------|----------|----|-------|
> | Flan-T5-XL 0-shot | **61.10** | 59.90 | 38.20 | 70.40 | 54.34 | 68.02 | **72.79** | 40.26 | 77.09 |
> | Flan-T5-XL Promptagator k-shot | 61.00 | 61.40 | 43.40 | 71.90 | 54.57 | 69.40 | 72.36 | **44.70** | **84.53** |
> | Flan-T5-XL DBS 1-shot | 59.80 | 62.50 | 44.10 | 72.00 | **54.81** | 69.42 | 72.69 | 44.07 | 83.66 |
> | Flan-T5-XL DBS 4-shot | 61.00 | **63.00** | **44.70** | **72.70** | 54.32 | **70.46** | 72.53 | 44.58 | 84.32 |
> |---------|-----|-----|------|---------|--------|-------|----------|----|-------|
> | Flan-T5-XXL 0-shot | 61.80 | 60.30 | 42.90 | 73.00 | 55.11 | **78.17** | 72.56 | 44.93 | 83.70 |
> | Flan-T5-XXL Promptagator k-shot | 61.90 | **63.30** | 47.40 | 73.80 | 55.32 | 78.00 | 73.53 | 47.90 | 85.56 |
> | Flan-T5-XXL DBS 1-shot | **62.40** | 62.80 | 47.60 | 74.30 | 55.41 | 77.62 | **74.11** | **48.46** | 85.31 |
> | Flan-T5-XXL DBS 4-shot | 62.30 | 62.30 | **47.70** | **74.50** | **55.71** | 77.68 | 73.78 | 48.41 | **85.73** |
> |---------|-----|-----|------|---------|--------|-------|----------|----|-------|
> | Palm 0-shot | 55.84 | 55.55 | 38.26 | 74.69 | 52.31 | 76.94 | 71.98 | 43.33 | 83.51 |
> | Palm Promptagator k-shot | 61.24 | 60.92 | **48.11** | **76.89** | **55.69** | 78.18 | **75.43** | 44.71 | **85.89** |
> | Palm DBS 1-shot | 58.50 | 60.62 | 46.99 | 74.87 | 53.43 | 78.07 | 72.93 | 42.91 | 85.52 |
> | Palm DBS 4-shot | **62.68** | **63.43** | 47.96 | 76.52 | 54.34 | **78.59** | 75.36 | **49.84** | 85.81 |

---

### Official Review · Reviewer_99xq · 2023-08-04

**Soundness:** 3

**Excitement:**

3: Ambivalent: It has merits (e.g., it reports state-of-the-art results, the idea is nice), but there are key weaknesses (e.g., it describes incremental work), and it can significantly benefit from another round of revision. However, I won't object to accepting it if my co-reviewers champion it.

**Missing References:**

1. Finding Supporting Examples for In-Context Learning
2. Active Example Selection for In-Context Learning

**Paper Topic And Main Contributions:**

This work investigates the use of LLMs for passage reranking. It proposes a difficulty-based demonstration selection strategy which is proven effective through experiments.

**Questions For The Authors:**

1. I do not think that it is a good idea for a short paper to claim that "To the best of our knowledge, we are the most extensive study to date investigating the use of demonstrations for LLM-based re-ranking."
2. If LLMs can generate answers directly, what is the point for them to rank passages?
3. Do you fine-tune LLMs?

**Reasons To Accept:**

1. This work focuses on an interesting question, which studies the capacity boundaries of LLMs in terms of passage reranking.

**Reasons To Reject:**

1. Many related works are missing and are not compared. For instance, works using LLMs for passage ranking, e.g., " Is chatgpt good at search? investigating large language models as re-ranking agent." and works proposing example selection for in-context learning, e.g., "Finding supporting examples for in-context learning."
2. There lacks in-depth analyses to prove why the proposed DBS works.
3. The contribution is not properly claimed.

**Reproducibility:**

3: Could reproduce the results with some difficulty. The settings of parameters are underspecified or subjectively determined; the training/evaluation data are not widely available.

**Reviewer Confidence:**

4: Quite sure. I tried to check the important points carefully. It's unlikely, though conceivable, that I missed something that should affect my ratings.

---

> ### Author Rebuttal · Authors · 2023-08-29
>
> Thank you for your thoughtful comments. We're glad to hear that our work is considered a simple and effective approach (E2cL,auEx) that improves a previous method (auEx), is helpful for future research (E2cL,auEx) and well motivated (auEx), and focuses on an interesting research question that pushes the boundaries of LLM capabilities (99xq). We respond to the questions from the review below.
>
> **Reason to Reject 1. Missing Related Work**
>
> Thank you for bringing these relevant papers to our attention. Although we did an extensive literature review, there was indeed a recent paper from arxiv that we failed to mention.
>
> We already cover Sun et al. [1] in our related work section. Demonstration selection is not a main focus of that work. Although they use a fixed set of demonstrations for relevance generation, they provide no criteria on how those demonstrations are selected. We will add additional clarification in our text.
>
> We already discussed Zhang et al. [2] in the related work section. This is an interesting paper, although it is for a different task, namely, sentence classification instead of passage re-ranking. It is inspired by active learning, but the method is quite different from ours and uses reinforcement learning. We will add additional clarification in our text.
>
> We are not aware of Li and Qiu [3] because it is on sentence classification instead of passage re-ranking, and it was publicly available on arxiv very recently (Feb 2023). Thank you for pointing it out and  we will add it in camera ready, and will also discuss the differences between demonstration selection for sentence classification vs. passage re-ranking.
>
> Both query and passage are important when selecting demonstrations for ranking tasks, and it involves two dimensions of complex text, making it different from the demonstration selection in sentence classification. It’s also worth noting that the uncertainty-inspired baselines in Li and Qiu are not particularly effective and often underperform random demonstrations and zero-shot sentence classification. DBS on the other hand consistently outperforms random demonstrations and zero-shot for passage re-ranking.
>
>
> [1] Sun et al., Apr 2023. https://arxiv.org/abs/2304.09542
>
> [2] Zhang et al., Nov 2022. https://arxiv.org/abs/2211.04486
>
> [3] Li and Qiu, Feb 2023. https://arxiv.org/abs/2302.13539
>
>
> **Reason to Reject 2. Lack of in-depth analysis**
>
> This is a very good question, and we try to answer it as follows.
>
> From a theoretical view, multiple recent studies suggest that in-context learning works similarly to gradient descent [1, 2], wherein the difficult training samples contribute more in terms of loss value. Empirically, we observe a positive correlation between the difficulty of demonstrations and the ranking performance when using such demonstrations in Figure 2. Thus, we believe such positive correlations are the reasons behind the success of DBS. We also report additional results on more dataset and using more models, and confirm the effectiveness of DBS (rebuttal to auEx).
>
> [1] https://arxiv.org/abs/2211.15661
>
> [2] https://arxiv.org/abs/2212.07677
>
>
> **Reason to Reject 3. Contributions and claim**
>
> Thank you for the suggestions. After investigating the literature, we tried to state the fact that this is the only work to date investigating the general methods to select demonstrations for LLM-based passage re-ranking. The most related works are [1] and [2]. The former investigates the LLM for re-ranking but with 0-shot configuration whereas the latter is to re-rank the document path for multihop QA. More recent papers like [3], [4], and [5] focus on the scoring method and are both under 0-shot setting, making it orthogonal to our method. We will rephrase the statement to make it more precise and clearer.
>
> [1] Sachan et al., Apr 2022. https://arxiv.org/abs/2204.07496
>
> [2] Khalifa et al., May 2022. https://arxiv.org/abs/2205.12650
>
> [3] Sun et al., Apr 2023. https://arxiv.org/abs/2304.09542
>
> [4] Ma et al., May 2023. https://arxiv.org/abs/2305.02156
>
> [5] Qin et al., June 2023. https://arxiv.org/abs/2306.17563
>
>
> **Q2. If LLMs can generate answers directly, what is the point for them to rank passages?**
>
> We agree that the LLM could generate answers. However, we argue that the ranking is still an important component in the QA system. For example, ranking could be used in the retrieval augmentation and in post-processing the generation results by sorting the generation candidates (e.g., remove aggressive generations). Beyond QA, ranking tasks are also widely used, for which Table 1 in the BEIR paper [1] includes several examples. Previous work [2] has shown that LLMs are effective re-rankers, and our work further investigates the use of demonstrations to further boost the ranking performance from LLM.
>
> [1] Thakur et al. https://arxiv.org/abs/2104.08663
>
> [2] Sachan et al., Apr 2022 https://arxiv.org/abs/2204.07496
>
>
> **Q3. Do you fine-tune LLMs?**
>
> No. We use the frozen LLM and investigate the demonstration selections.

---

### Official Review · Reviewer_auEx · 2023-08-10

**Soundness:** 2

**Excitement:**

3: Ambivalent: It has merits (e.g., it reports state-of-the-art results, the idea is nice), but there are key weaknesses (e.g., it describes incremental work), and it can significantly benefit from another round of revision. However, I won't object to accepting it if my co-reviewers champion it.

**Paper Topic And Main Contributions:**

This work builds upon a previous retrieval reranking method that uses LLMs with zero-shot (no examples to help the LLM). It demonstrates how adding an example (1-shot approach) can improve re-ranking performance. The proposed method, PaRaDe, utilizes few-shot examples that are included in the prompt during the reranking process. To effectively add more examples, the work also proposes that the selection of such examples be done through a "difficulty-based demonstration" selection strategy. This strategy selects challenging demonstrations which is defined as selecting the query-document pair with the lowest likelihood from training samples as the few-shot examples.


**Questions For The Authors:**

Question A: Low likelihood can also mean that the query and demonstration are distantly or even not related. What kind of preprocessing is done to insure that the query-document pair with the lowest likelihood are still related?

**Reasons To Accept:**

1. Investigates how few-shot paradigm would work in a retrieval setting that also works to improve a previous method.
2. The work provides a simple and accessible method that avoids finetuning and is likely extensible to a variety of retrieval settings.
3. Shows that there is high variance when using few-shot demonstrations selected at random, this serves as a motivation for the proposed Difficulty-based Selection strategy.

**Reasons To Reject:**

1. While the author shows that (1) there is high variance when using randomly selected demonstrations and that (2) using demonstrations selected through the proposed strategy results in a score improvement, the variance when using Difficulty-based Selection strategy is not shown. Both compared methods still show improvement over zero-shot and the motivating reason to utilize Difficulty-based Selection strategy over random selection would then be that it is more consitant.
2. It is never stated why FLAN-T5-XL/XXL was used in this work. When referring to the query likelihood retrieval model, it does seem that decoder-only models could work just as well. Which leaves out out whether this approach only works for a specific model architecture or not.

**Reproducibility:**

4: Could mostly reproduce the results, but there may be some variation because of sample variance or minor variations in their interpretation of the protocol or method.

**Reviewer Confidence:**

3: Pretty sure, but there's a chance I missed something. Although I have a good feel for this area in general, I did not carefully check the paper's details, e.g., the math, experimental design, or novelty.

**Typos Grammar Style And Presentation Improvements:**

Description of the LLM Re-ranking by Query Likelihood could be more elaborate to show that the query q is a sequence of q_i:q_N (or a sequence of N tokens) and that for a collection of documents D, each document d \in D to clarify that d is in fact a subset of D. The proposed Difficulty-based Selection that uses DQL could also be elaborated with a variation of the provided equations to make it clearer what it is for the readers.

---

> ### Author Rebuttal · Authors · 2023-08-29
>
> Thank you for your thoughtful comments. We're glad to hear that our work is considered a simple and effective approach (E2cL,auEx) that improves a previous method (auEx), is helpful for future research (E2cL,auEx) and well motivated (auEx), and focuses on an interesting research question that pushes the boundaries of LLM capabilities (99xq). We respond to the questions from the review below.
>
> **Reason to Reject 1. What is the variance of DBS?**
>
> Thank you for pointing this out! DBS is deterministic given the pool of demonstrations, so there is no variance for the selection of demonstration (or variance is 0); BM25 and 0-shot are also deterministic. Rather, we look at the relative benefit of DBS compared to random demonstration selection. We show that DBS outperforms or is on-par with the average of random selection. Importantly, DBS avoids the low performance observed with some random demonstrations.
>
> For random selection we use 30x 1-shot random selected demonstrations from the training data, then evaluated on a 1000 query subset of the test data and reported the mean, standard deviation, min, and max performance across random 1-shot demonstrations. Due to the expensiveness of the random experiments, so far we have only used FLAN-T5-XXL and 4 of the datasets. We will include the full data in our next revision. There are already extensive experiments using random demonstrations and TRECDL-2020 in our initial submission.
>
> |  | FEVER | HotpotQA | NQ | Quora |
> |---------|-------|----------|----|-------|
> | No. of Queries | 1000 | 1000 | 1000 | 1000 |
> | BM25 | 75.3 | 60.3 | 32.8 | 78.8 |
> | UPR 0-shot | 80.3 | 70.7 | 47.0 | 82.5 |
> | DBS 1-shot (Ours) | **81.7** | **72.7** | 48.4 | **86.6** |
> | Random 1-shot Avg. / Std. | 80.2 / 1.49 | 71.7 / 0.25 | **48.7** / 0.6 | 84.8 / 0.37 |
> | Random 1-shot Min. | 77.3 | 71.2 | 47.2 | 83.9 |
> | Random 1-shot Max. | 83.2 | 72.1 | 49.6 | 85.5 |
>
> **Reason to Reject 2. What is the motivation for using FLAN-T5-XL/XXL?**
>
> Our motivation for using FLAN-based models is that they are open source (and hence reproducible) models that are known to have in-context learning capabilities [1]. Since its release, it has been verified to have strong few-shot prompting performance [2]. We use FLAN-T5 models as a generalization of LLM in this work. To further confirm the effectiveness of DBS, we include results from Palm in the following. We also include results on five more datasets that have annotated training data available (BioASQ, FEVER, HotpotQA, NQ, Quora).
>
> [1] https://arxiv.org/abs/2210.11416
>
> [2] https://arxiv.org/abs/2210.02406
>
> |   | T19 | T20 | FiQA | Scifact | BioASQ | FEVER | HotpotQA | NQ | Quora |
> |---------|-----|-----|------|---------|--------|-------|----------|----|-------|
> | BM25 | 50.60 | 48.00 | 23.60 | 66.50 | 46.45 | 75.32 | 60.27 | 32.84 | 78.83 |
> |---------|-----|-----|------|---------|--------|-------|----------|----|-------|
> | Flan-T5-XL 0-shot | **61.10** | 59.90 | 38.20 | 70.40 | 54.34 | 68.02 | **72.79** | 40.26 | 77.09 |
> | Flan-T5-XL Promptagator k-shot | 61.00 | 61.40 | 43.40 | 71.90 | 54.57 | 69.40 | 72.36 | **44.70** | **84.53** |
> | Flan-T5-XL DBS 1-shot | 59.80 | 62.50 | 44.10 | 72.00 | **54.81** | 69.42 | 72.69 | 44.07 | 83.66 |
> | Flan-T5-XL DBS 4-shot | 61.00 | **63.00** | **44.70** | **72.70** | 54.32 | **70.46** | 72.53 | 44.58 | 84.32 |
> |---------|-----|-----|------|---------|--------|-------|----------|----|-------|
> | Flan-T5-XXL 0-shot | 61.80 | 60.30 | 42.90 | 73.00 | 55.11 | **78.17** | 72.56 | 44.93 | 83.70 |
> | Flan-T5-XXL Promptagator k-shot | 61.90 | **63.30** | 47.40 | 73.80 | 55.32 | 78.00 | 73.53 | 47.90 | 85.56 |
> | Flan-T5-XXL DBS 1-shot | **62.40** | 62.80 | 47.60 | 74.30 | 55.41 | 77.62 | **74.11** | **48.46** | 85.31 |
> | Flan-T5-XXL DBS 4-shot | 62.30 | 62.30 | **47.70** | **74.50** | **55.71** | 77.68 | 73.78 | 48.41 | **85.73** |
> |---------|-----|-----|------|---------|--------|-------|----------|----|-------|
> | Palm 0-shot | 55.84 | 55.55 | 38.26 | 74.69 | 52.31 | 76.94 | 71.98 | 43.33 | 83.51 |
> | Palm Promptagator k-shot | 61.24 | 60.92 | **48.11** | **76.89** | **55.69** | 78.18 | **75.43** | 44.71 | **85.89** |
> | Palm DBS 1-shot | 58.50 | 60.62 | 46.99 | 74.87 | 53.43 | 78.07 | 72.93 | 42.91 | 85.52 |
> | Palm DBS 4-shot | **62.68** | **63.43** | 47.96 | 76.52 | 54.34 | **78.59** | 75.36 | **49.84** | 85.81 |
>
>
> **Question A: Low likelihood can also mean that the query and demonstration are distantly or even not related. What kind of preprocessing is done to ensure that the query-document pair with the lowest likelihood are still related?**
>
> We perform manual filtering on the top demonstrations (chosen by DBS) to remove the (query, passage) pairs that are wrongly labeled in the training data, according to the respective definitions for individual dataset. This is also mentioned in Footnote 5: “We remove the pairs that are labeled incorrectly. The demonstrations used are included in Appendix A.3.1.” Note that this manual filtering serves to correct the annotation errors from the training dataset and requires minimal manual effort.
>
> In particular, on the training dataset, we exploit FLAN-T5-XXL to score each positive (query, passage) pair using QL and sort them in ascending order. We go through the list and skip the (q, d) pairs that are with wrong labels. We stop after four demonstrations are collected. We follow the annotation guideline from the respective dataset. Here are the ranks of the selected demonstrations from the ascending list and the number of (q, d) pairs skipped for each dataset after removing the duplicates query-passage pairs. For FEVER, we only consider pairs that are with “support” annotation as demonstrations, whereas BEIR converts both “refute” and “support” as positive samples. This makes the skipped (q, d) pairs on FEVER relatively high, which is still below 20.
>
> |   | Ranks of the selected demos | #(q,d) skipped |
> | ---|------------|------------------ |
> | MSMarco | 1,2,3,4 | 0 |
> | FiQA | 4,5,6,7 | 3 |
> | Scifact | 5,6,7,8 | 3 |
> | BioASQ | 1,3,5,18 | 5 |
> | FEVER | 5,8,16,22 | 17 |
> | HotpotQA | 2,3,6,8 | 4 |
> | NQ | 3,4,5,8 | 4 |
> | Quora | 1,2,4,5 | 1 |
>
> **Typo and grammar style**
>
> Thank you for the suggestion! We will clean up the descriptions and notations of DBS in the camera ready version.

---

### Meta-Review · Area_Chair_dAFv · 2023-09-14

**Recommendation:** 4

**Metareview:**

The paper proposes and approach to improve LLM-based re-ranking performance by including even a single example in the prompt. This improvement makes a significant impact under specific conditions that a detailed analysis investigates.

The paper is sound and well written. The authors were very effective, in my view, in motivating and explaining the value of the approach proposed (see details in the reviews and rebuttals). I think the paper will make an interesting contribution to the conference.

---

### Decision · Program_Chairs · 2023-10-07

**Decision:**

Accept-Findings

**Comment:**

The paper proposes and approach to improve LLM-based re-ranking performance by including even a single example in the prompt. This improvement makes a significant impact under specific conditions that a detailed analysis investigates.

The paper is sound and well written. The authors were very effective, in my view, in motivating and explaining the value of the approach proposed (see details in the reviews and rebuttals). I think the paper will make an interesting contribution to the conference.